# Role of NMDA Receptor in High-Pressure Neurological Syndrome and Hyperbaric Oxygen Toxicity

**DOI:** 10.3390/biom13121786

**Published:** 2023-12-13

**Authors:** Alice Bliznyuk, Yoram Grossman

**Affiliations:** 1Ilse Katz Institute for Nanoscale Science and Technology (IKI), Ben-Gurion University of the Negev, Beer Sheva 8410501, Israel; 2Department of Physiology and Cell Biology, Zlotowski Center for Neuroscience, Faculty of Health Sciences, Ben-Gurion University of the Negev, Beer-Sheva 8410501, Israel; ramig@bgu.ac.il

**Keywords:** NMDAR, HPNS, HBOTx, high pressure, GluN2A, Zn^2+^ voltage-independent inhibition

## Abstract

Professional divers exposed to pressures greater than 11 ATA (1.1 MPa) may suffer from high-pressure neurological syndrome (HPNS). Divers who use closed-circuit breathing apparatus and patients and medical attendants undergoing hyperbaric oxygen therapy (HBOT) face the risk of CNS hyperbaric oxygen toxicity (HBOTx) at oxygen pressure above 2 ATA (0.2 MPa). Both syndromes are characterized by reversible CNS hyperexcitability, accompanied by cognitive and motor deficits, and N-methyl-D-aspartate receptor (NMDAR) plays a crucial role in provoking them. Various NMDAR subtypes respond differently under hyperbaric conditions. The augmented currents observed only in NMDAR containing GluN2A subunit increase glutamatergic synaptic activity and cause dendritic hyperexcitability and abnormal neuronal activity. Removal of the resting Zn^2+^ voltage-independent inhibition exerted by GluN2A present in the NMDAR is the major candidate for the mechanism underlying the increase in receptor conductance. Therefore, this process should be the main target for future research aiming at developing neuroprotection against HPNS and HBOTx.

## 1. Introduction

Pressure, like temperature, is one of the fundamental physical factors affecting living organisms. Most humans are physiologically adapted to live at an ambient atmospheric pressure of 1 ATA (the average atmospheric pressure exerted at sea level, 0.1 MPa). However, some are exposed to pressures as low as 0.3 ATA (0.03 MPa) on the summit of Mt. Everest at an altitude of 8848 m or as high as 70 ATA (7.09 MPa) at a depth of 690 m under the ocean surface [1]. Although only relatively few highly trained personnel are exposed to these extreme environments, many others frequently encounter levels of hyperbaric pressures (HP, i.e., more than 1 ATA) on a more regular basis, such as divers using an underwater breathing apparatus (SCUBA) for recreational, professional (for oil and salvage industry), and combat purposes; patients and medical attendants undergoing hyperbaric oxygen therapy (HBOT); and working in the compressed atmosphere of a subterranean environment such as tunneling [2].

Organisms exposed to HP develop behavioral and physiological changes such as hyperexcitability, tremors, myoclonus, and complete seizures. The compression rate governs the type of adaptation mechanisms available in response to HP, which only cope with relatively slow rates. It has been shown that, for humans, a compression of 50–60 ATA would require between 5 and 6 days to return to a fully functioning state [3]. Moreover, the adaptation process is not linear—the higher the pressure, the longer adaptation takes at a steady state. For any given species, there is a pressure beyond which adaptation cannot compensate for pressure effects. Approximately 1000 ATA (10 km sea water) appears to be the ultimate limit for the feasibility of any complex life form since higher pressures would dictate protein coagulation and total inhibition of enzymatic activity. It seems that the less complex a life form with a less complex nervous system, the more easily it adapts to HP.

On the other hand, over 20 species of marine mammals routinely dive to extreme depths, ranging between 600 and over 2000 m (60–200 ATA) for as long as 20 up to 120 min [4,5,6,7]. Evidently, these species have evolved modifications to critical physiological processes that allow their body in general, and specifically their nervous system, to withstand these great pressures.

Professional divers may suffer from direct HP effects that present many physiological challenges, primarily affecting the lungs, hollow viscera, and nervous system. The nature of the breathing gas is also an important factor. Air-breathing animals and humans exposed to O_2_ partial pressure above 2 ATA may suffer from hyperbaric oxygen toxicity (HBOTx) (see below), and N_2_ partial pressure above 5 ATA induces N_2_ narcosis (inert-gas narcosis) as well as CO_2_ toxicity. Most of these neurological impairments can be alleviated and even eliminated by controlling the partial pressures of absorbed tissue gases at normal values while under HP conditions using various devices. Yet, there is still one limitation for professional deep divers exposed to pressures greater than 11 ATA: High Pressure Neurological Syndrome (HPNS) [8,9,10,11]. This leaves HPNS as the last major limiting factor for confronting the HP environment and deep diving. HPNS is characterized by reversible central nervous system (CNS) hyperexcitability and cognitive and motor deficits. Symptoms of HPNS include tremors, myoclonic jerking [12], somnolence, EGG changes, visual disturbance, nausea, dizziness, and decreased mental performance [13,14]. HPNS susceptibility and symptoms intensity depend on the compression rate and the absolute ambient pressure. It is conceivable that this constellation of signs and symptoms originates in disturbances in the synaptic activity of neuronal networks (for review, see [15]). In addition to HPNS, prolonged repetitive deep-sea operations of professional divers at HP may result in permanent memory and motor impairment [16].

Previous studies have demonstrated that the hyperexcitability of HPNS is induced mainly by N-methyl-D-aspartate receptors (NMDARs) [17,18,19,20,21,22]. In contrast, other members of the iGluR family play a very small part in inducing CNS hyperexcitation at HP. The AMPA receptor did not respond significantly to HP [23], and kainate receptors were unaffected by pressure [24]. Other amino acid-activated ionotropic receptors, such as GABA receptors, are also insensitive to HP [25]. The maximal response of glycine receptors remained unchanged, although the IC50 was considerably elevated at HP [24].

CNS HBOTx is a risk for recreational, commercial, and combat divers who use closed-circuit breathing apparatus (100% O_2_ or a mixture of O_2_ and one or more inert gases) at O_2_ partial pressures greater than 2 ATA. Major symptoms of HBOTx are CNS hyperexcitability, convulsions, and loss of consciousness, which are usually reversible [26,27,28,29]. Similar CNS symptoms may occur among patients and medical attendants undergoing HBOT. It is commonly accepted that the general mechanism underlying HBOTx is probably oxidative stress, namely the presence of high levels of reactive oxygen species (ROS) that may damage any human biological system [29]. However, recently, we have shown that HPNS and HBOTx may share a common mechanism in which NMDAR plays a crucial role [30] (see below).

## 2. NMDAR Subtypes and Their Response to Pressure

The NMDAR belongs to the family of ionotropic glutamate receptors that mediate the majority of excitatory neuronal transmission within the CNS [31,32]. Three families of NMDAR subunits have been identified [33,34,35]: GluN1 contains eight distinct isoforms (GluN1-1a to 4a and GluN1-1b to 4b) owing to RNA splicing [36]; the GluN2 family is composed of four members (GluN2A to D) encoded by four different genes; GluN3 subunits (GluN3A and GluN3B) arise from two separate genes. NMDARs function as heterotetrameric assemblies (see Figure 1), usually associating two GluN1 and two GluN2 subunits (GluN1⁄GluN2 complexes) or a mixture of GluN1, GluN2, and GluN3 subunits [32,37,38]. NMDARs incorporating GluN3 subunits are thought to form either di-heteromeric (GluN1⁄GluN3) or tri-heteromeric (GluN1⁄GluN2⁄GluN3) complexes [32]. A large repertoire of homologous NMDAR subunits allows for various combinations of subunit assembly, giving rise to 32 receptor subtypes in the CNS [39].

To date, there are abundant but incomplete data on the NMDAR subtypes’ spatial distribution and function(s) in the mammalian brain [31,37,40,41,42]. GluN2A and GluN2B are the predominant subunits in the adult CNS, particularly in higher brain structures (such as the hippocampus and cortex) [41,43,44], indicating that they have central roles in synaptic function and plasticity. In addition, in the hippocampus and cortex, tri-heteromeric GluN1/GluN2A/GluN2B receptors also populate, with estimates of abundance ranging from 15% to >50% of the total receptor population [45,46,47]. Tri-heteromeric GluN1/GluN2A/GluN2C and GluN1/GluN2B/GluN2D receptors have also been described [37,38].

NMDAR has been implicated with CNS hyperexcitability as part of HPNS. Neuropharmacological studies at HP have suggested increased NMDAR responses in CA1 pyramidal cells [17,18,19,20,21,22]. In the same brain region, electrophysiological studies showed a significant increase in the synaptic NMDAR response followed by postsynaptic excitability changes [48,49]. The first attempt to directly measure the NMDAR currents at HP was made by Daniels and his colleagues (1998), showing that HP increased the receptors’ currents expressed in Xenopus laevis oocytes.

Electrophysiological studies from our laboratory [50] have revealed differential current responses under HP He (obtaining HP pressure with He gas) conditions in NMDAR subtypes that contain either GluN1-1a or GluN1-1b splice variants co-expressed in Xenopus laevis oocytes, with all four GluN2 subunits (Table 1). These discoveries, particularly regarding the above-detailed ‘pair’, suggest that the difference in HP He response may be related to the small difference between the a and b variants of the GluN1 receptor.

Further study from our lab in which six GluN1 splice variants (GluN1-1a/b, GluN1-2a/b, and GluN1-3a/b) were co-expressed with the GluN2A subunit (which is most abundant in adult brains and CNS and plays a crucial role in long-term potentiation (LTP) and learning) showed a different amount of increase in the ionic currents at HP He [51]. The control current amplitudes and the relative increase under HP He conditions differed and depended on the GluN1 variant present in the subtype analyzed. In addition, the GluN1-4a/b splice variant co-expressed with GluN2A/B exhibited “dichotomic” (either increased or decreased) responses at HP He [52]. The study on GluN1 variants and previous studies on GluN2 subunits [15,22,47,48,49,52] indicate a very complex picture of NMDAR response under HP He conditions. Current evidence indicates that this augmentation is considered one of the key elements causing HPNS and possibly long-term irreversible CNS impairment. Studying the HP He modulation of specific NMDAR subtypes revealed important information about their function in different brain areas and even in specific neuron types.

A recent intriguing study from our lab [30] on the GluN1-1a co-expressed with GluN2A/B showed at HBO (obtaining HP pressure with 2–6 ATA oxygen gas) the same response as under HP He, namely, an increase in the GluN1-1a + Glu2A subtype response and no change in the GluN1-1a+GluN2B subtype response. This led us to the idea of a possible common mechanism for both responses (see below).

## 3. The Mechanism of the NMDAR Hyperexcitation

The GluN2A and GluN2B subunits are the most abundant in the adult CNS and generate large currents. Electrophysiological studies discussed above showed no change or reduction in the current of GluN1 + GluN2B/2C/2D receptors under HP He (6/7 subtypes, Table 1) and HBO. In contrast, GluN1 + GluN2A receptor responses increased by up to 63% at HP He (6/8 subtypes, Table 1) and up to 30% at HBO. We therefore suggest that GluN2A is the only subunit responsible for the CNS hyperexcitability induced by HP He and probably by HBO (see below).

NMDAR is a unique ligand-gated ionotropic receptor (see Figure 1): binding of both agonist glutamate and co-agonist glycine (or d-serine) to their recognition sites on the receptor subunit assembly is required for receptor activation [37]. NMDAR channels are highly Ca^2+^-permeable in addition to Na^+^ and K^+^. They display unusually slow kinetics deactivation due to slow glutamate unbinding. These usually enable a considerable increase of [Ca^2+^]_i_, triggering many neuronal modulation mechanisms. NMDAR has various blockers, modulators, and antagonists (see [31] for their detailed list and site of action).

### 3.1. Voltage-Dependent Mg^2+^ and Zn^2+^ Inhibition

Di-heteromeric receptors that contain GluN2A or GluN2B are well known to have high voltage-dependent sensitivity to mM range [Mg^2+^]_o_ blockade, i.e., strong membrane depolarization (induced via an alternative pathway) can remove this inhibition. The Mg^2+^ binding site (containing Asn residues, N and N C 1 sites) is located at the channel pore in the trans-membrane domain (TMD, see Figure 1D) of the receptor [53,54]. Mg^2+^ voltage-dependent sensitivity is controlled by a single GluN2 residue in the M3 segment [55] significantly affecting the relative contribution of NMDAR subtypes to synaptic integration and plasticity. Similarly, [Zn^2+^]_o_ in the range of [mM] can inhibit NMDARs through a voltage-dependent channel block that appears to occur within the ion channel pore and may involve residues (e.g., Asn616 in the M2 region of GluN1 and Asn614/5 on GluN2) that are known to be associated with other divalent ions blockade [35,56,57,58,59,60].

It is unlikely that the dynamic changes in the NMDAR responses at HP He or HBO may occur due to such large changes in the divalent ions’ concentration. However, long-term modulation was suggested as a result of some change in the Mg^2+^ sensitivity of the block [22].

### 3.2. NMDAR Characteristics: Affinity, Stoichiometry, Surface Expression

A change in NMDAR affinity to glutamate under HP He conditions should also be considered [61]. The kainate-type receptor was slightly potentiated (<14%) but showed no change in EC50 [24]. The NMDAR current was markedly increased (128%), but EC50 was not reported. In contrast, HP did not affect the GABAA receptor current and EC50, whereas, while the maximal current of the Glycine receptor was not affected, its EC50 increased by 60% [24]. In those studies, the researchers used polyA+ mRNA extracted from rat brain tissues without knowing the exact subunit combinations. We confirmed using specific cRNA in three NMDAR subtypes that the glutamate dose–response curve and EC50 are not affected by HP He [51]. In addition, our investigations [30,51,52,62,63,64] have demonstrated that aggregate formation, modified stoichiometry, and reversal potential alterations cannot explain the increase in the current of NMDAR containing the GluN2A subunit at HP He.

During LTP, additional NMDARs are inserted into the membrane [65]. In our experiments, a similar process may occur during the 15–20 min time required for HP stabilization. The immunoprecipitation experiment carried out in our laboratory indicated that there is no increase in surface expression of NMDARs in the oocyte membrane under HP He conditions [63]. It is conceivable to conclude that there are no HP-induced modifications in NMDAR affinity to glutamate, stoichiometry, aggregate formation, its reversal potential, and surface expression that could explain the above-mentioned increase in the response.

### 3.3. Structural Differences of the GluN1 Variants

GluN2A co-expressed with different splice variants of GluN1 showed different currents at normobaric and HP He and depended on the GluN1 variant (Table 1) [51]. Our measurements showed that GluN1-1a and GluN1-2a mean current amplitudes saw a greater increase due to HP He than their corresponding “b” splice variants (see [51] Figure 5A). In contrast, for GluN1-3a and GluN1-4a, the opposite behavior was observed. This may indicate that there might be a possible functional relevance to the existence of the exon 5-encoded 21 amino acids extracellular N1 loop [36] (in the N terminal domain (NTD)), which is present in “b” types of GluN1 variants and absent in “a” types. However, there are other sequence differences between the GluN1 variants which are located in the intracellular C terminal domain (CTD) and result from differential splicing of exons 21 and 22 [66]. GluN1-1 splice variants have the longest CTD due to the expression of both exon 21 and the long portion of exon 22. GluN1-2 variants do not express exon 21 but have the same long exon 22 as GluN1-1. GluN1-3 and GluN1-4 variants have much shorter CTDs since only a small portion of exon 22 is present, and only GluN1-3 contains exon 21. The CTD region is known to be involved in receptor trafficking and signaling. From our experiments, we may postulate that CTDs are also involved in current modulation. In general, the observed effect of HP He-induced increased/decreased inward ionic currents through the different NMDAR subtypes can be sufficiently explained by observing the increase/decrease in input conductance (the slope of the I/V curve of each variant) of the oocyte under HP conditions (see [52] Figure 2B). The good correlation between these two parameters may support the ‘single switch’ hypothesis for HP ‘channel’ opening.

### 3.4. Direct HP Effects: Molecular Dynamic Simulation (MDS)

The mechanism(s) underlying NMDAR conductance increase/decrease is still not clear. Unfortunately, at this stage of the research, we can only speculate about possible explanations since we cannot analyze experimentally a single receptor function. Therefore, the use of protein structure modeling together with advanced molecular dynamic simulation (MDS) may yield new insights and suggest original directions for HP physiology research. In our first attempt to use MDS [64] to distinguish between HP He and pressure per se (hydrostatic) effects, we simulated a single NMDAR protein (GluN1-1a + GluN2B, whose sequences and 3D structure were available at that time) embedded in a small patch of single phospholipid (DOPC) bilayer, with artificial periodic boundary conditions. In addition, the CTD domains were missing, and no membrane potential could be applied. Earlier experimental studies indicated that HP may affect the protein structure directly or indirectly by modifying membrane properties such as fluidity, structure, and net volume. Such changes are induced mainly through a decrease in lateral spacing of the phospholipids [67], while the acyl chains become straighter [68]. We postulated that such modifications in the membrane should impose a tertiary conformation change in the trans-membrane domain (TMD) of the receptor, which consequently changes its properties. Our major findings indicate that hydrostatic pressure and HP He have different impacts on the cell membrane and the receptor tertiary structure. Analyses of various statistical and specific model parameters indicated that HP He generally induced more distortions and instability on the receptor structure than HP. The clearest effect is the increased volume of the lipid membrane due to He diffusion into the hydrophobic core of the DOPC, creating “micro-bubbles” between the two layers. The detailed He and water solvent layouts along NMDAR segments show significant redistribution of channel solvation patterns (sort of “dehydration”) that may directly affect the protein pore regions (see Figure 1). HP He is much more effective in reducing the DOPC acyl chains stability factor, causing spatial distortion. Simulations of both compression methods revealed that the pore diameter is reduced to the level of being considered ‘close state’ (NMDAR lost its agonists), yet HP He is less effective, and the pore surface showed much more conservation of the tertiary structure than the “open state”. Another important region of the TMD is the pore Mg^2+^ binding site (see Section 3.1). Both HP and HP He increased the distances between the four known binding sites of the Mg^2+^ (on Asn residues of the M2 region of the TMD) and was associated with the distorted alignment of the M4 peripheral TMD α-helix segment, which is critical for NMDAR activation and desensitization. These may reduce, to some extent, the efficacy of the Mg^2+^ block [22]. The MDS clearly suggest that HP He cannot be considered as pure HP effects on the NMDARs. It is important to note that the effects of other inert noble gases could be very different. The implications of this hypothesis are further discussed in Section 4.

### 3.5. Voltage-Independent Zn^2+^ Inhibition

The fact that GluN2A (but not other members of the GluN2 family) has harbors for Zn^2+^ ions on its NTD that can act in a voltage-independent manner to reduce the frequency of channel opening has been known for more than a decade. The concentration of Zn^2+^ needed for that is <5 nM [56,57,58,66]. It was only a few years ago that we noticed that this voltage independent Zn^2+^ inhibition exists only in the GluN2A subunit, which is the only subtype of NMDAR that showed a significant increase of the current under HP He and HBO conditions. Taking this observation together with our previous negative results, we postulated that the HP-induced current increase of NMDARs containing the GluN2A subunit may partially result from the removal of the voltage-independent Zn^2+^ inhibition of the receptor. Although all experiments discussed above were carried out in nominally zero Zn^2+^, contamination of Zn^2+^ in the nM range was probably present, which means that the Zn^2+^ inhibition site of GluN2A could be occupied. Indeed, a study from our lab [30] showed that the elimination of Zn^2+^ by TPEN (a selective powerful Zn^2+^ chelator) at normobaric pressure increased the NMDAR current by 50%, revealing the presence of strong resting voltage-independent Zn^2+^ inhibition of the GluN1-1a + GluN2A receptor. Most importantly, the addition of TPEN under both HP He and HBO conditions failed to alter the currents. The lack of any additional response to TPEN strongly supports the notion that voltage-independent inhibition by Zn^2+^ had already been removed under both conditions. The absence of any effect of HP He and HBO on the GluN1-1a + GluN2B NMDAR, which lacks the Zn^2+^ voltage-independent inhibition site, greatly strengthens this hypothesis. However, the increase in current at 5.4 ATA HBO was only 35%, whereas at 51 ATA HP He was 63%. It appears that the removal of Zn^2+^ inhibition via HBO is less effective than HP He, or there is some additional component in the response to the HP He increase in currents.

## 4. Clinical Aspects and Considerations

HPNS symptoms are usually reversible upon decompression (see introduction). An intriguing Norwegian report [16] has suggested that repetitive exposure to HP over years may cause chronic memory and motor impairments in professional divers. The HP He that they may have been exposed to was either just sub-threshold to HPNS or “supra-threshold”, but HPNS symptoms were antagonized by the use of narcotic gas mixtures (such as Trimix containing O_2_, N_2_, and He). We hypothesize that even if clear HPNS symptoms were not observed, the glutamate NMDAR response was still potentiated, causing more Ca^2+^ flow into the neurons. Through several signal transduction pathways, an overload of Ca^2+^ can activate metabolic cascades that deteriorate the neuron and eventually may lead to cell death via apoptosis [67]. Therefore, the long-term health effects are not separate phenomena but rather an accumulation of minute deleterious changes inflicted by the potentiation of NMDARs during each deep dive. Thus, this represents a permanent consequence of at least part of the wide symptoms and signs of HPNS.

Our MDS study [64] suggested that hydrostatic pressure and compression with He have different impacts on the cell membrane and the tertiary receptor structure. Professional divers who showed HPNS symptoms usually perform their dive with a gas mixture that contains He. Taking this together with our MDS results, we can speculate that the presence of the He in diver breathing mixtures could partly contribute to HPNS symptoms. Another fact that may support this hypothesis is the ability of some marine mammals to perform very deep breath-hold dives when they are exposed only to hydrostatic pressure [62]. Our MDS study shows that pressure per se has a less devastating influence on the membrane and the embedded protein, which may render their CNS less vulnerable to ambient HP.

## 5. Conclusions

We generally assume that HPNS and HBOTx result from the dysfunction of neuronal network synaptic activity and that NMDAR plays a crucial (but not sole) role in provoking it.The augmented currents only in NMDAR containing GluN2A subunit increase glutamatergic synaptic activity and cause dendritic hyperexcitability and abnormal neuronal activity.Aggregate formation, modified stoichiometry, alterations in glutamate and glycine affinity, increase in receptor expression, and change in reversal potential cannot explain the increase in current of the GluN2A subunit.Voltage-dependent divalent ions inhibition has a limited role in the HP He and HBO effects.The preexisting structural modifications in GluN1 variants may slightly modulate the NMDARs response, which is determined using the GluN2 subtype.HP He-induced increased/decreased inward ionic currents (GluN2A included) can be sufficiently explained by observing the increase/decrease in input conductance (pore permeability) of the receptor.MDS suggests that hydrostatic pressure and compression with He may cause direct alterations in NMDAR protein conformation and cell membrane properties. However, they may have different impacts which are not fully understood.Removal of the resting Zn^2+^ voltage-independent inhibition exerted by GluN2A present in the NMDAR is the major candidate for the mechanism underlying the increase in the receptor conductance.The clinical aspects of HPNS and HBOTx should be reexamined in light of the new molecular and modeling findings dealing with compressed gases.New methods for neural protection should be explored in light of recent discoveries, especially the role of Zn^2+^ voltage-independent inhibition in the HP response.Despite immense technological progress and great computational capabilities advancement, we remain limited in reaching the ocean bottom. If we ever want to explore those areas, pressure susceptibility must be studied to remove restrictions that prevent us from entering the deep ocean frontiers.

## Figures and Tables

**Figure 1 biomolecules-13-01786-f001:**
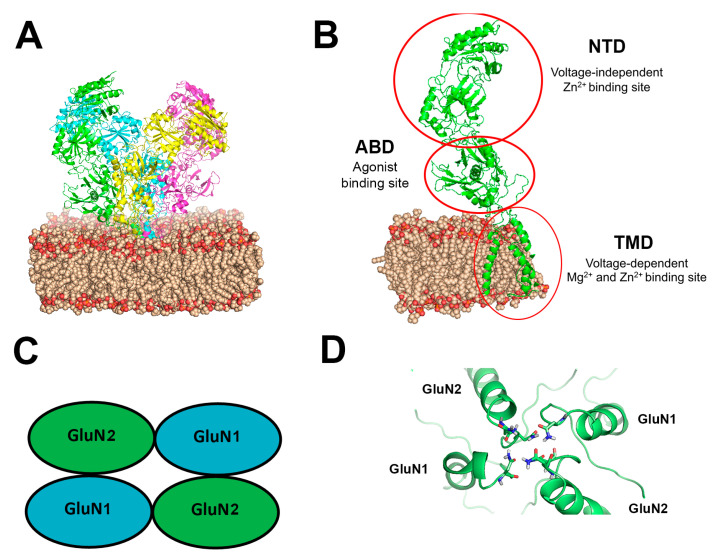
(**A**) Simulated NMDAR embedded in a DOPC lipid bilayer at 1 ATA pressure. The NMDAR is shown as a cartoon colored with chains of the four subunits (green, magenta, yellow, cyan), DOPC is presented as spheres in wheat and red color (colored by element), (**B**) One subunit of the NMDAR in the membrane, important sites indicated with a red circle. All GluN subunits share a modular architecture that is composed of four distinct domains: the N-terminal domain (NTD), the agonist-binding domain (ABD) that binds glycine or d-serine in GluN1 and GluN3 and glutamate in GluN2, the transmembrane domain (TMD) containing the ion channel, and an intracellular C-terminal domain (CTD) (not shown in the figure). The NTD and CTD are the most divergent regions. (**C**) Schematic topology of the NMDAR and its subunits (**D**) NMDAR asparagine (Asn) residues coordinations at the Mg^2+^ site. Asn residues are shown as sticks colored by element on the M2 region of the TMD of the four subunits.

**Table 1 biomolecules-13-01786-t001:** Summary of the different subtypes of the NMDAR response to 50 ATA HP He. Receptor current responses under HP He conditions in NMDAR subtypes that contain GluN1 splice variants co-expressed with all four GluN2 subunits in Xenopus laevis oocytes and measured using a two-electrode voltage clamp technique. NT—not tested, =—no change, ↑—increase, ↓—decrease, ↕—dichotomy, %—averaged data, SDs were omitted for clarity.

GluN1 Splice Variant	GluN2A	GluN2B	GluN2C	GluN2D
GluN1-1a	↑ (63%)	=	↓ (−25%)	=
GluN1-1b	↑ (25%)	=	↓ (−47%)	↓ (−24%)
GluN1-2a	↑ (45%)	NT	NT	NT
GluN1-2b	↑ (32%)	NT	NT	NT
GluN1-3a	↑ (28%)	NT	NT	NT
GluN1-3b	↑ (42%)	NT	NT	NT
GluN1-4a	↕ (26%) (−26%)	↕ (8%) (−14%)	NT	NT
GluN1-4b	↕ (39%) (−43%)	NT	NT	NT

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
