# Peer review of "Role of NMDA Receptor in High-Pressure Neurological Syndrome and Hyperbaric Oxygen Toxicity"

_biomolecules, 2023, doi:10.3390/biom13121786_

Round 1
Reviewer 1 Report
Comments and Suggestions for Authors
The manuscript by A. Bliznyuk and Y. Grossman is a review of the effects of (mainly) high pressure of Helium (such as encountered in human deep saturation diving) and of hyperbaric oxygen on the mammalian NMDA receptors. It would have been easier to understand the manuscript without having to read the publication if the authors could have given more detailed information about the data from the previous studies used in the review instead of only indicating their reference. Nevertheless, the manuscript is well written, thank you for giving me opportunity to review it. I have some minor comments and questions.
Some references are lacking: lines 97, 134, 166, 179, 236 and 263.
Part 3.3: the first paragraph (lines 213-234) is duplicated (lines 235-256).
I am not sure to understand correctly if the data cited in the manuscript were obtained with dimeric assemblies or tetrameric structures with only two subunits subtypes. Could you please clarify this ?
I understand that the review is focused on the effects of two gases (Helium and Oxygen). However, are there data concerning the effects of other hyperbaric gases on the NMDA receptors which could add to the discussion?
In the paragraph 3.4, the hypothesis that high helium pressure could modify NMDAR function through changes of the membrane characteristics is evoked. Data obtained at atmospheric pressure and which describe the influence of membrane environment on the function of NMDARs could add to the discussion. Does such data exist ?
Is the reference 53 (Bliznyuk et al. The Mechanism of NMDA Receptor Hyperexcitation in High Pressure Helium and Hyperbaric Oxygen. Front Physiol 2020, 11, 1057) the only study focused on the effect of HBO on NMDARs ? Or on the glutamate neurotransmission in the brain ? If not, are there other similarities between HP He and HBO ? Such data would reinforce the hypothesis evoked in the manuscript that hyperexcitability induced by these two conditions share common mechanisms.
Author Response
The manuscript by A. Bliznyuk and Y. Grossman is a review of the effects of (mainly) high pressure of Helium (such as encountered in human deep saturation diving) and of hyperbaric oxygen on the mammalian NMDA receptors. It would have been easier to understand the manuscript without having to read the publication if the authors could have given more detailed information about the data from the previous studies used in the review instead of only indicating their reference. Nevertheless, the manuscript is well written, thank you for giving me opportunity to review it. I have some minor comments and questions.
During our research efforts in recent years, we obtained many negative answers to our relevant working hypotheses. These are briefly listed in the text (including their references) for the purpose of concision. However, positive findings are described in detail.
Some references are lacking: lines 97, 134, 166, 179, 236 and 263.
Line 97- the references for all sentences indicated in the end of the sentence [31,33,36]
Line 134- the reference was indicated in the line 131, [49]
Line 166 (new number 174),236 (new number 223) and 263 (section 3.4) – the references were added
Line 179 (new number186)- the reference was indicated in the line 185, [55]
Part 3.3: the first paragraph (lines 213-234) is duplicated (lines 235-256).
Corrected: the duplicated lines were omitted.
I am not sure to understand correctly if the data cited in the manuscript were obtained with dimeric assemblies or tetrameric structures with only two subunits subtypes. Could you please clarify this?
The functional NMDARs are always heterotetrameric, usually containing two subunits of GluN1 and two subunits of GluN2. This is clearly stated in line 100 and illustrated in Fig 1 A,C,D. We added "of the four subunits" in the legends of Fig1A to stress the point of tetrameric nature, The topology of the receptor is clearly indicated in Fig1C and its spatial tetrameric nature in Fig1D.
I understand that the review is focused on the effects of two gases (Helium and Oxygen). However, are there data concerning the effects of other hyperbaric gases on the NMDA receptors which could add to the discussion?
To the best of our knowledge, no experimental data are available on the response of NMDARs exposed to pressure with other gases. As we indicated in the Introduction lines 54-72, mammals exposed to pressurized air may also suffer from N2 narcosis and CO2 toxicity. We are currently working on another MDS project in which we studied the NMDAR response to noble gases such as neon, argon and xenon under normal and pressure conditions. We expect some differences between the gases' effects due to their atomic size, but presently, it is too early for any definitive conclusions. Therefore, we do not think that the addition of this issue to the discussion is helpful.
In the paragraph 3.4, the hypothesis that high helium pressure could modify NMDAR function through changes of the membrane characteristics is evoked. Data obtained at atmospheric pressure and which describe the influence of membrane environment on the function of NMDARs could add to the discussion. Does such data exist ?
Paragraph 3.4 was extensively rewritten to answer this question.
Is the reference 53 (Bliznyuk et al. The Mechanism of NMDA Receptor Hyperexcitation in High Pressure Helium and Hyperbaric Oxygen. Front Physiol 2020, 11, 1057) the only study focused on the effect of HBO on NMDARs ? Or on the glutamate neurotransmission in the brain ? If not, are there other similarities between HP He and HBO ? Such data would reinforce the hypothesis evoked in the manuscript that hyperexcitability induced by these two conditions share common mechanisms.
As far as we know, we were the only group that measured directly NMDAR response under HBO conditions. We are unfamiliar with research that specifically studied the response of AMPA, GABA and kainite receptors under HBO conditions as were performed with He HP (we referred to it in the introduction). In our article [30] and this review, we focus specifically on the mechanism in which NMDAR increases its currents under HP He and HBO conditions and its role in inducing HPNS and HBOTx.
Reviewer 2 Report
Comments and Suggestions for Authors
The authors have reviewed studies, including their own research, on the NMDAR containing GluN2A subunit and its postulated role in the neural hyperexcitability and abnormal activity that occurs when exposed to hyperbaric Helium (HP He) and hyperbaric oxygen (HBO). The former occurs at very high pressures (>11 ATA pressure per se, once corrected for hyperoxia and inert gas narcosis of nitrogen) and is known as High Pressure Nervous Syndrome (HPNS). The latter is known as CNS oxygen toxicity (HBOTx), which occurs at >2 ATA O2. The authors propose that removal of the resting Zn2+ voltage-independent inhibition exerted by GluN2A present in the NMDAR is the major candidate for the mechanism underlying the increase in receptor conductance during exposure to HP and HBO and that future research should focus on this target for developing neuroprotection against HPNS and HBOTx.
The manuscript is well organized and written, and it is very useful in that it outlines several lines of research going forward for those studying HPNS and HBOTx. However, the manuscript would benefit from additional details including alternative explanations for their observations and hypotheses. Specific items to consider for inclusion/revision are as follows:
1) I agree with their 9th Conclusion (lines 344-345) that “The clinical aspects of HPNS and HBOTx should be reexamined in light of the new molecular and modeling findings dealing with compressed gases.” However, some of their conclusions are drawn quite strongly without considering other possibilities. For example, the authors state emphatically in the Abstract that “…this process should be the main target for future research aiming at developing neuroprotection against HPNS and HBOTx.” I agree that this is the case for the NMDAR subunit in HPNS; however, I’m not entirely convinced that it is the “main target” in HBOTx, for a couple of reasons. The authors postulate that GluN2A is the only subunit responsible for the increased excitability caused by exposure to HP He and HBO (lines 164-165). Their stimuli, from what I can gather by looking at their research paper, was 5 ATA Oxygen (HBOTx) and 50 ATA Helium (HPNS). Do we know if 5 ATA He without hyperoxia stimulated GluN2A? Perhaps the HBO sensitivity was actually pressure sensitivity to hyperbaric He. Recall that 2-4 ATA of HP He can stimulate brainstem neurons (see Mulkey et al 2003 J. Appl. Physiol.). In addition…
2) …measurements of GLUT and GABA in the brain during exposure to HBO shows that GLUT does not increase but that GABA decreases prior to seizure (see: Zhang et al. Brain Research Protocols (2005) 14: 61-66). This, in turn, decreases the GABA:GLUT ratio and increases excitability. Moreover, Gaiser et al report that it is Inhibition of glutamic acid decarboxylase (GAD) during HBO exposure that decreases GABA and thus inhibition leading to increased excitability. They propose that S-nitrosylation of GAD65 is implicated in decreased GAD activity and oxygen-induced seizures (see Gasier et al. (2017) Neurosci. Letts. 653: 283-287). These data support the alternative hypothesis that it is GABA and not GLUT that is an important target for HBOTx. Can the authors please reconcile their hypothesis with these data.
3) lines 268-269. The authors state that “MDS also suggests that hydrostatic pressure and compression with He have different impacts on the cell membrane and the receptor tertiary structure.” The manuscript would benefit for the uninitiated of a short discussion of the use of HP He vs. hydrostatic pressure when studying HPNS and barosensitivity at lower levels of pressure.
4) Text associated with Table 1 should include a brief description of cell preparation used.
5) There are multiple error messages regarding references in the text that need to be addressed (see Error! Reference source # not found.)
6) line 89. Reference 9 seems to be the wrong reference based on the text.
Author Response
The authors have reviewed studies, including their own research, on the NMDAR containing GluN2A subunit and its postulated role in the neural hyperexcitability and abnormal activity that occurs when exposed to hyperbaric Helium (HP He) and hyperbaric oxygen (HBO). The former occurs at very high pressures (>11 ATA pressure per se, once corrected for hyperoxia and inert gas narcosis of nitrogen) and is known as High Pressure Nervous Syndrome (HPNS). The latter is known as CNS oxygen toxicity (HBOTx), which occurs at >2 ATA O2. The authors propose that removal of the resting Zn2+ voltage-independent inhibition exerted by GluN2A present in the NMDAR is the major candidate for the mechanism underlying the increase in receptor conductance during exposure to HP and HBO and that future research should focus on this target for developing neuroprotection against HPNS and HBOTx.
The manuscript is well organized and written, and it is very useful in that it outlines several lines of research going forward for those studying HPNS and HBOTx. However, the manuscript would benefit from additional details including alternative explanations for their observations and hypotheses. Specific items to consider for inclusion/revision are as follows:
1)I agree with their 9th Conclusion (lines 344-345) that “The clinical aspects of HPNS and HBOTx should be reexamined in light of the new molecular and modeling findings dealing with compressed gases.” However, some of their conclusions are drawn quite strongly without considering other possibilities. For example, the authors state emphatically in the Abstract that “…this process should be the main target for future research aiming at developing neuroprotection against HPNS and HBOTx.” I agree that this is the case for the NMDAR subunit in HPNS; however, I’m not entirely convinced that it is the “main target” in HBOTx, for a couple of reasons. The authors postulate that GluN2A is the only subunit responsible for the increased excitability caused by exposure to HP He and HBO (lines 164-165). Their stimuli, from what I can gather by looking at their research paper, was 5 ATA Oxygen (HBOTx) and 50 ATA Helium (HPNS). Do we know if 5 ATA He without hyperoxia stimulated GluN2A? Perhaps the HBO sensitivity was actually pressure sensitivity to hyperbaric He. Recall that 2-4 ATA of HP He can stimulate brainstem neurons (see Mulkey et al 2003 J. Appl. Physiol.). In addition…
We are well aware of the report on some respiratory medullar neurons that are sensitive to 2-4 ATA HP He. In fact, the results, among others from the same laboratory, are described in the review cited by us (Dean et al J. Appl. Physiol 2003). Yet, this is quite a rare finding, in contrast to numerous reports dealing with HPNS indicating that the threshold for HPNS in animals as well as many other in vitro preparations (tested using submerged tissue method) is in the range of 10-15 ATA. Furthermore, even in the above-mentioned review, the authors describe in Fig 6 a couple of cells from the medullar solitary tract and hippocampal slice that "were determined to be baro-insensitive during compression with hyperbaric He". In addition, in an older study from our laboratory (Trasiuk and Grossman J Appl Physiol 1991) on the function of the respiratory system in isolated rat newborn medulla-spinal cord preparation, we found no evidence for any such high sensitivity to low He pressure. The threshold for HPNS-measured parameters such as spontaneous motor neuron's rhythmic activity, AP burst parameters, medulla-spinal cord and spinal reflexes, were above 20 ATA. The experiments were performed at pressures up to 100 ATA, and even upon decompression to 2-4 ATA, all values returned to the control level. The preliminary experiments for most of the data described in our manuscript included compression to lower levels in order to determine the most effective range for the experiments. Usually, we had to compress any of the NMDAR subtypes expressed in the oocytes to levels higher than 20-25 ATA in order to detect a significant change in the currents. Therefore, it is quite unlikely that GluN2A is sensitive to low pressure of 5 ATA. Another important point is that HBO above 2 ATA can trigger HBOTx symptoms. But the pressure per-ce is not mandatory for that since any chemical oxidants or high level of ROS can cause HBOTx.
"Probably" was inserted in line 171
2) …measurements of GLUT and GABA in the brain during exposure to HBO shows that GLUT does not increase but that GABA decreases prior to seizure (see: Zhang et al. Brain Research Protocols (2005) 14: 61-66). This, in turn, decreases the GABA:GLUT ratio and increases excitability. Moreover, Gaiser et al report that it is Inhibition of glutamic acid decarboxylase (GAD) during HBO exposure that decreases GABA and thus inhibition leading to increased excitability. They propose that S-nitrosylation of GAD65 is implicated in decreased GAD activity and oxygen-induced seizures (see Gasier et al. (2017) Neurosci. Letts. 653: 283-287). These data support the alternative hypothesis that it is GABA and not GLUT that is an important target for HBOTx. Can the authors please reconcile their hypothesis with these data.
There is no doubt that reduced inhibitory (mostly GABAergic) synaptic activity is a viable hypothesis for HBoTx (or HPNS) mechanism. Decreased synthesis of GABA will certainly cause reduced presynaptic release. However, the extracellular concentrations of GABA or glutamate in the brain dialysates is not the best way to demonstrate that. These probably indicate neurotransmitters release and /or uptake mechanisms rather than the actual efficacy of synaptic transmission. For example, if the glutamate release is unchanged or even decreased, it may be compensated for by an increased current of the receptor (probably NMDAR and not other glutamate receptors, see introduction to HPNS). The final outcome of the synaptic activity could be further modulated or even amplified by changes in Ca channels activity in the dendritic membrane. Indeed, in many invertebrate peripheral and mammalian central brain synapses the actual excitatory and inhibitory synaptic responses are impaired under pressure and HBO conditions (see review articles). The present review presents our recent exciting findings suggesting a new molecular target (GluN2A) for HPNS and possibly HBOTx. This new avenue of research is as viable as any former research hypotheses that so far have not yielded any neural protection for those syndromes.
"(but not sole)" was inserted into conclusion 1.
3) lines 268-269. The authors state that “MDS also suggests that hydrostatic pressure and compression with He have different impacts on the cell membrane and the receptor tertiary structure.” The manuscript would benefit for the uninitiated of a short discussion of the use of HP He vs. hydrostatic pressure when studying HPNS and barosensitivity at lower levels of pressure.
Paragraph 3.4 was extensively rewritten to answer this question.
Additional aspects of this question are dealt with in par 4, lines 316-325 (new paragraph 3.4 points this out).
4) Text associated with Table 1 should include a brief description of cell preparation used.
Added in fig 1 legend as suggested.
5) There are multiple error messages regarding references in the text that need to be addressed (see Error! Reference source # not found.)
We could not see these messages in the text. We made a special effort to eliminate such errors.
6) line 89. Reference 9 seems to be the wrong reference based on the text.
The reference was replaced.